# The Effects of Cortical Reorganization and Applications of Functional Near-Infrared Spectroscopy in Deaf People and Cochlear Implant Users

**DOI:** 10.3390/brainsci12091150

**Published:** 2022-08-28

**Authors:** Xiaoqing Zhou, Menglong Feng, Yaqin Hu, Chanyuan Zhang, Qingling Zhang, Xiaoqin Luo, Wei Yuan

**Affiliations:** 1Department of Otolaryngolgy, Chongqing General Hospital, Chongqing 401147, China; 2Chongqing Medical University, Chongqing 400042, China; 3Chongqing School, University of Chinese Academy of Sciences, Chongqing 400714, China; 4Chongqing Institute of Green and Intelligent Technology, University of Chinese Academy of Sciences, Chongqing 400714, China

**Keywords:** cochlear implant, speech perception, cross-modal reorganization, functional near-infrared spectroscopy

## Abstract

A cochlear implant (CI) is currently the only FDA-approved biomedical device that can restore hearing for the majority of patients with severe-to-profound sensorineural hearing loss (SNHL). While prelingually and postlingually deaf individuals benefit substantially from CI, the outcomes after implantation vary greatly. Numerous studies have attempted to study the variables that affect CI outcomes, including the personal characteristics of CI candidates, environmental variables, and device-related variables. Up to 80% of the results remained unexplainable because all these variables could only roughly predict auditory performance with a CI. Brain structure/function differences after hearing deprivation, that is, cortical reorganization, has gradually attracted the attention of neuroscientists. The cross-modal reorganization in the auditory cortex following deafness is thought to be a key factor in the success of CI. In recent years, the adaptive and maladaptive effects of this reorganization on CI rehabilitation have been argued because the neural mechanisms of how this reorganization impacts CI learning and rehabilitation have not been revealed. Due to the lack of brain processes describing how this plasticity affects CI learning and rehabilitation, the adaptive and deleterious consequences of this reorganization on CI outcomes have recently been the subject of debate. This review describes the evidence for different roles of cross-modal reorganization in CI performance and attempts to explore the possible reasons. Additionally, understanding the core influencing mechanism requires taking into account the cortical changes from deafness to hearing restoration. However, methodological issues have restricted longitudinal research on cortical function in CI. Functional near-infrared spectroscopy (fNIRS) has been increasingly used for the study of brain function and language assessment in CI because of its unique advantages, which are considered to have great potential. Here, we review studies on auditory cortex reorganization in deaf patients and CI recipients, and then we try to illustrate the feasibility of fNIRS as a neuroimaging tool in predicting and assessing speech performance in CI recipients. Here, we review research on the cross-modal reorganization of the auditory cortex in deaf patients and CI recipients and seek to demonstrate the viability of using fNIRS as a neuroimaging technique to predict and evaluate speech function in CI recipients.

## 1. Introduction

According to the Global Burden of Disease (GBD) study, deafness is the fourth most prevalent disabling disease in the world [1,2]. The estimates from the World Health Organization (WHO) show that hearing loss affects 466 million people worldwide, of whom 93% are adults and 7% are children [3,4], and that number will rise to almost 900 million by 2050 [4]. Additionally, the yearly cost of hearing-related disorders throughout the world is around USD 750 billion [4]. A hearing impairment dramatically reduces patients’ quality of life and places a heavy strain on society and families. In children, hearing loss impacts not only the development of speech, language, and cognitive skills but also psychological health. Therefore, preventing and treating deafness is a significant societal issue that we are currently facing.

CI is a neuroprosthetic device that uses biomedical engineering technology to bypass the cochlea and directly deliver electrical stimulation to the auditory nerve [5]. People with severe-to-profound SNHL will see a considerable improvement in their quality of life, auditory detection, and speech perception skills after undergoing CI surgery and extensive auditory rehabilitation training. To date, CI has grown to be both the most popular and most effective method of treating total sensory loss [6]. Over 736,900 deaf people worldwide are currently receiving CI, up from over 12,000 in 1995, according to the most recent figures (NIDCD 2019). However, despite the optimal conditions of patients and the best efforts of clinicians, there is still significant variation in speech perception abilities among CI recipients [7,8]. The fair expectations of patients and their families are severely constrained by this variability, which also affects their choice of whether or not to undergo CI.

In fact, variability in CI outcomes can be attributed to multiple factors. The common influencing factors include duration of hearing deprivation [8,9]; age at onset of deafness [8]; natural age at CI [10]; use of hearing aid (HA) [11]; side of CI [12,13]; depth of electrode implantation [14]; residual hearing [11,15]; cochlear nerve integrity [16,17]; and neurocognitive function [10,18]. Only up to 20% of the variability could be accounted for by modeling based on these variables [11]. Unfortunately, there are currently no reliable predictors of auditory performance in CI. Behavioral tests like questionnaires and the speech perception score/threshold test are presently the mainstay methods used to assess speech performance in CIs. These tests’ outcomes are susceptible to patient, audiometrist, and testing material bias and may be unreliable, especially for infants and children. Obviously, an objective instrument for evaluating speech understanding would have significant clinical utility and be more efficient and reliable among different CI individuals.

It may be valuable to examine the brain changes from deafness to hearing restoration and the central processing involved in understanding auditory speech with a CI [19,20,21,22]. Then, combining these neuromarkers with behavioral measures is more accurate and effective for guiding post-implant programming, modifying rehabilitation training strategies following CI, assessing speech and language outcomes, and eventually predicting CI outcomes before implantation. In this review, we emphasize the importance of cortical factors, namely the auditory cortex reorganization. We next examine the evidence for this reorganization and how it relates to CI performance, as well as the rationale behind why it might predict CI variability. Additionally, we work to clarify the viability of using fNIRS as a neuroimaging tool to predict and evaluate speech function in CI recipients.

## 2. Role of Cross-Modal Reorganization in the Auditory Cortex

### 2.1. Evidence for Cross-Modal Reorganization in the Auditory Cortex

In adapting to internal and external influences during growth and adulthood, the cerebral cortex consistently changes in response to sensory input, insult, damage, and learning. When sensory input is abnormal, cross-modal reorganization, a type of cortical neuroplasticity, takes place. This reorganization leads to the cortex with deprived modality becoming vulnerable to intact sensory modalities [20]. Researchers generally believe that the deprivation of specific sensory function gradually results in the atrophy of cortical representations (such as function or metabolism). This phenomenon may not be completely reversible even after the original sensory function is restored, which ultimately affects the rehabilitation performance. According to research, hearing deprivation in deafness alters the connection between higher-order neurocognitive centers, the auditory system, and other sensory systems as well as within the auditory system due to cross-modal and intra-modal changes [20,23]. Among them, the cross-modal reorganization between the auditory system and other sensory systems is considered to be one of the most important changes. Cats with congenital deafness, for instance, exhibited improved peripheral localization and visual motion detection abilities, which was thought to be related to the posterior auditory field activity [24]. Patients with severe-to-profound hearing loss outperform normal-hearing individuals in recognizing communicative gestures [25], speech reading ability [26,27], and the neural processing of visual signals in auditory cortex [28]. For those with hearing loss, the auditory cortex is always recruited for different visual inputs such as visual motion [25,29,30,31,32,33,34], biological motion [35,36,37], silent speech reading [26,38,39,40,41], peripheral visual stimuli [42,43], and non-sign-related hand shapes [44] as well as for the linguistic processing of sign language [26,44,45,46,47,48,49,50] and for different somatosensory inputs [42,51,52,53,54]. Similar sensation-enhancing behaviors have also been demonstrated in people with CI [31,55,56,57,58,59].

Deaf people mostly communicate with others through visual language. After auditory deprivation or degeneration, dependence on visual language signals will lead to cross-modal reorganization by vision in the auditory cortex. The auditory cortex may also be engaged in cross-modal reorganization by somatosensory inputs due to the proximity of the auditory cortex to the somatosensory cortex and the overlap between subcortical neurons in response to respective signals. This review focuses on the cross-modal reorganization by vision because visual language signals are crucial in natural and complicated communication.

Visual cross-modal reorganization provides superior visual functions to compensate for hearing deficits in deafness, identified as an adaptive effect [24]. On the other hand, this reorganization is considered to be a key factor affecting hearing outcomes after auditory recovery (e.g., CI), and the dominant view holds that these two are negatively correlated, that is, that cross-modal reorganization is detrimental to auditory performance after implantation [58,60]. This conventional viewpoint, however, appears to be oversimplified and constrained in recent years [61,62]. For example, some scholars argue that visual language response in the auditory cortical area during deafness could maintain the typical development of language networks and promote hearing and speech rehabilitation following CI rather than limiting auditory function restoration [62,63,64]. These two conflicting viewpoints suggest that despite the fact that cross-modal plasticity is a significant component influencing CI outcomes, further research is necessary to fully understand the relationship between the two. By comparing and analyzing the similarities and differences among existing research findings, we hope to uncover possible reasons for the contradictory relationship between cross-modal reorganization and hearing performance following implantation. Table 1 summarizes the main studies on the correlates between cross-modal reorganization and CI outcomes, including authors, year, study population, stimuli, neuroimaging method, main results, and study design.

### 2.2. Maladaptive Plasticity Effects in the Auditory Cortex

How will the auditory input that has been restored interact with the cross-modal reorganization driven by auditory deprivation prior to CI? In fact, data from human and animal experiments suggest that the cross-modal recruitment of auditory brain regions by visual stimuli interferes with the processing of auditory stimuli in the auditory cortex [65,66]. In 2001, Lee et al. used multiple regression analysis to determine the relationships between the duration of hearing loss, the use of CI, and metabolic activity as predictors of hearing recovery. They chose PET–CT for monitoring the metabolism in the auditory cortex of 15 prelingually deaf children both before and after CI. The findings demonstrated that the metabolic activity of the auditory cortex could predict speech recognition in an independent and powerful manner. In addition, the low metabolic activity of this cortex (bilateral superior temporal gyrus) was favorably connected with speech performance following CI [60]. Following this, Hyo-Jeong Lee et al. [67] and Oh et al. [68] also found that the auditory performance of CI recipients decreased with increasing glucose metabolic activity of the auditory cortex in the resting state prior to the intervention. However, the last two studies had small sample sizes (11 and 3 deaf children, respectively), and thus, the findings should be interpreted with caution. In a PET imaging study by Hyo-Jeong Lee et al. (2007) [69], who examined the association between cortical metabolic activity before implantation and speech intelligibility three years after implantation among 22 prelingually deaf children, it was discovered that the metabolic activity of the right superior temporal gyrus increased with CI use and was adversely correlated with speech results. According to the authors’ analysis, the increased glucose metabolism meant that other sensory modalities had taken over the auditory cortex, preventing the recovery of auditory function after CI and reflecting the unfavorable impact of the cross-modal reorganization of the auditory cortex on auditory recovery. In accordance with earlier PET investigations, EEG studies also discovered that speech rehabilitation with CI was inversely proportionate to the cross-modal recruitment of auditory brain regions by visual stimuli [58,66,70,71]. The effect, however, was mostly seen in the right hemisphere, and it was measured in terms of either N1 amplitude or, in other instances, N1 latency [70,72,73]. In conclusion, the current CI rehabilitation strategy is to avoid the use of visual language to maintain the ability of auditory brain regions to process auditory language, thus optimizing the success of CI.

The deleterious effect of cross-modal reorganization with CI may occur for one of three reasons: (1). In cases of congenital or early-onset deafness, the dendritic branch structure is interrupted, and the auditory cortex’s stimulation pathway structure is unpruned because of hearing deprivation, which gives other sensory modalities the chance to utilize the auditory brain areas [64,74,75]. The auditory recovery following a CI is constrained because the newly acquired auditory stimulation must compete with other signals for brain resources [75]. (2). Another possible explanation is a functional decoupling between the primary cortex and the higher cortex after hearing absence. An undamaged auditory system is made up of a dense network of bottom-up and top-down connections that ensures information comparison and allows for computing a prediction error (the difference between expected and actual input), a signal that originates and pushes adult learning [65,76,77,78,79]. The importance of top-down connections in auditory processing increases with aging, enabling brain networks to generalize their responses and to preserve associated neural response patterns. However, when cross-modal reorganization takes place in the auditory cortex, functional decoupling between the bottom-up and top-down connections is triggered [80,81], which ultimately prevents the top-down connections from developing normally and the full recovery of the auditory system functioning [80,82]. (3). Due to resource competition brought on by cross-modal plasticity, there are not enough areas of the auditory cortex that can process auditory information.

### 2.3. Adaptive Plasticity Effects in the Auditory Cortex

Recent research suggests that, whether or not individuals have experience with audio-visual language, cross-modal reorganization of the auditory cortex by visual stimuli is likely advantageous to the development of auditory speech perception following CI [62,64,67,69,83,84,85]. A retrospective study was conducted to evaluate CI outcomes between early implanted deaf children with deaf parents and children with hearing parents at various points following implantation. The findings indicated that children from deaf families had superior language and speech perception to that of children from normal families. The participants in both groups were matched based on onset and severity of deafness; duration of deafness; age at CI; duration of CI; gender; and CI model. The preliminary results indicated that early contact with sign language and early implantation are beneficial to audiological rehabilitation after implantation, emphasizing the exposure to visual language for maintaining the linguistic systems in deaf children and highlighting the potential adaptive effects of cross-modal reorganization on sensation recovery [85]. In addition, Anderson et al. [62] used fNIRS to investigate the relationship between CI success and the cross-modal activation of auditory brain areas by visual speech in 17 deaf people from before to 6 months after implantation. The findings demonstrated that the visual takeover of the auditory cortex had adaptive benefits for CI users since higher visual language-activated auditory cortex responses were linked to greater speech comprehension. It is noted that there was heterogeneity in subjects, who included prelingually, perilingually, and postlingually deaf individuals, so that incorporating these subjects into a unified framework of analysis requires care. Several studies conducted in blind individuals consistently showed that the mechanisms of cross-modal reorganization differ between individuals with early and late sensory deprivation [86,87,88] and in the latter are highly influenced by age at onset of blindness [87]. Nevertheless, this result is in line with a recent observation in an animal model that revealed that responses to CI auditory stimuli are not excluded by the cross-modal reorganization of auditory brain areas; therefore, this reorganization should not be strictly regarded as maladaptive for CI outcomes [89]. Mushtaq et al. [64] also applied fNIRS to examine the impact of cross-modal plasticity in the bilateral superior temporal cortex (STC) on speech understanding in 19 children with CI and 20 normal children under 4 conditions, and they discovered that the STC of CI children was activated by visual language while that of normal children was not, even though there was no difference in the activation of auditory language between the groups. The authors thought that visual language cross-activating STC did not come at the expense of cortical sensitivity to auditory stimuli; instead, visual and auditory language had a synergistic effect, suggesting a positive influence of audiovisual speech on hearing restoration with CI. However, since only 3 of the 19 CI children in this study performed poorly on the phonetic perception test, statistical analyses could not be conducted between CI children with good and poor perception due to a lack of statistical power. Despite the drawbacks of these studies, it is undeniable that they all discovered adaptive effects of cross-modal reorganization on speech ability following CI implantation, which highlights the shortcomings and limitations of the conventional view and demands additional research in this area.

There may be two reasons for the adaptive effects of CI in some individuals: (1) Since auditory language and visual language representation are inherently related, the activation of the superior temporal cortex by visual language may be a reflection of inner language and auditory imagery processes [90]. In this way, this type of cross-modal activation may imply a stronger correspondence or synergistic impact between the two modalities, ultimately aiding in the recovery of auditory function. In fact, speech perception can be improved with the multimodal integration of auditory and visual language signals, and CI users are more adept at this skill than people with normal hearing [91]. (2) Animal model studies from the past can provide detailed information on the neurophysiological consequences of congenital and neonatal deafness [63]. However, unlike with CI users, these models only concentrate on the development of the auditory system rather than that of the language network. It is evident that understanding speech involves both healthy linguistic development and normal auditory function, but language cannot be interpreted using an animal model [63]. Strong visual speech activation in the temporal cortex, particularly in the left superior temporal cortex in CI recipients, may indicate that the language network has developed favorably throughout the critical period and that the cortical language processing circuits have matured, ultimately enabling better speech production by interpreting auditory data through audiovisual mechanisms [62,64,92].

**Table 1 brainsci-12-01150-t001:** Design information and key results from the main studies on the effects of cortical plasticity.

Manuscripts	Study Population	Stimuli	Neuroimaging Method	Main Results	Study Design
Lee D.S et al., 2001 [60]	Prelingually deaf patients (2.2–20.3 y), *n* = 15	No, resting state	PET-CT	Low metabolic activity of the auditory cortex was positively correlated with CI outcomes (r = 0.81).	Longitudinal study
Doucet M E et al., 2006 [71]	Prelingually and postlingually deafened CI users (18–62 y), *n* = 13	Visual stimuli consisted of a high-contrast sinusoidal concentric grating.	EEG	The poor CI performers exhibited broader, anteriorly distributed, high P2 amplitudes over the cortex, whereas the good CI performers showed significantly higher P2 amplitudes over visual occipital areas.	Cross-sectional
Lee H.J et al., 2007 [69]	Prelingually deaf patients (1.5–11.3 y), *n* = 22	No, resting state	PET-CT	Decreased metabolic activity in the right Heschl’s gyrus (R = −0.45) and in the posterior superior temporal sulcus (R = −0.466) were positively correlated with CI outcomes.	Longitudinal study
Buckley K A et al., 2011 [70]	Prelingually and postlingually deafened CI users (14–65 y), *n* = 22	Visual stimuli consisted of moving visual gradients with still pictures of cartoon characters in the center.	EEG	A clear negative association between the amplitude of the N1 VEP over the right temporal lobe and speech perception scores was observed for prelingually deafened CI users (r = −0.7703 to −0.8965) but not for postlingually deafened CI users.	Cross-sectional
Sandmann P et al., 2012 [58]	Postlingually deafened CI users (38–69 y), *n* = 11	Visual stimuli consisted of reversing displays of chequerboard patterns.	EEG	At the P100 latency, CI users showed activation in the right auditory cortex that was inversely related to speech recognition ability in a CI recipient.	Cross-sectional
Campbell J et al., 2016 [73]	Prelingually and postlingually deafened CI users (4.95–15.43 y), *n* = 14	Visual stimuli consisted of a high-contrast sinusoidal concentric grating.	EEG	The VEP N1 latency in the right temporal cortex was negatively related to speech perception in background noise in children with cochlear implants (r = −0.576).	Cross-sectional
Liang M et al., 2017 [72]	Pre-lingually deafened CI children (4.2–6.4 y), *n* = 20	Visual stimuli consisted of a photograph with imaginative sound and a photograph without imaginative sound.	EEG	Good CI performers showed significant decreases in N1 amplitude in the primary auditory cortex and in the primary visual cortex, but these did not occur in the poor CI performer group.	Longitudinal study
Anderson et al., 2017 [62]	Prelingually, perilingually, and postlingually deaf adults (36–78 y), *n* = 15	IHR number sentences (normal speech, male and female speakers) were split into visual speech (visual-only or lip-reading) and auditory speech (auditory-only)	fNIRS_ETG4000	There was a strong positive correlation between the change in bilateral STC activation to visual speech from preimplantation to postimplantation and speech understanding with a CI (r = 0.70).	Longitudinal study
Anderson et al., 2019 [93]	Prelingually, perilingually, and postlingually deaf adults (36–78 y), *n* = 15	IHR number sentences (normal speech, male and female speakers) were split into visual stimuli (visual-only or lip-reading) and auditory stimuli (auditory-only)	fNIRS_ETG4000	Although stronger activation to visual speech preoperatively was predictive of poorer speech understanding outcomes postimplantation (r =−0.75), this relationship was driven by the heterogeneity in subjects.	Longitudinal study
Mushtaq et al., 2020 [64]	Prelingually deaf children (6–11 y), *n* = 19	Visual speech, auditory speech, signal correlated noise, and steady speech shaped noise.	fNIRS_ETG4000	Visual and auditory speech are processed synergistically in the temporal cortex of children with CIs, and they should be encouraged, rather than discouraged, to use visual speech.	Cross-sectional

### 2.4. Summary of Existing Evidence

Two essential phases are necessary for the development of a particular cortical region: The first is associated with the maturation of sensory pathways, and the second is associated with the development of specific functions in which a given cortical region is engaged [61]. These two key phases for the auditory cortex include the formation of connections within auditory processing (hearing function) and the formation of distinct cognitive functions (language function) that are intimately tied to the auditory cortex [61]. Visual take-over of the auditory cortex after hearing deprivation could promote the development of language function in the critical period, which may be beneficial to the prognosis following CI. However, it should be emphasized that not all types of cross-modal reorganization in the auditory cortex are necessarily adaptive to audiological rehabilitation after implantation; in fact, only those associated with language network development may be useful.

Additionally, hearing subjects also showed significant response to both auditory and visual speech stimuli in STC, which was consistent with the multi-modal tendency [38,94]. Speech reading is widely believed to induce activity in the bilateral STC [26,95], with a higher level of activity in the left STC than that in the right [39,40,90]. There is abundant evidence that the left hemisphere is specialized in language processing regardless of the pattern of sensory input, such as auditory input for oral communication and visual input for speech reading or sign language [44,48,96,97], and therefore, maintaining the specialization of left hemisphere auditory language processing may be an important cortical factor in the success of CI [98]. Lazard et al. observed that strong responses to vision–language stimuli in the left STC were associated with a better CI outcome, whereas cortical responses in the right STC before implantation were associated with a poorer CI outcome [99]. The reason may be that the right hemisphere tends to handle information independent of the linguistic network [30,44], which seems to be consistent with results that cross-modal plasticity in the right temporal cortex is detrimental to auditory recovery [70,72,73]. Therefore, it could be assumed that the cross-modal reorganization of the left STC by visual speech during hearing deprivation helps to maintain the left hemisphere specialization, facilitating postoperative recovery after CI implantation.

The majority of studies on the unfavorable effects of cross-modal plasticity in the auditory cortex on the prognosis for CI use electroencephalogram (EEG), PET, and computed tomography (PET–CT) to assess cortical changes. Despite being compatible with CI, the resting state of the auditory cortex’s metabolic activity under PET–CT lacks specificity. It is unknown whether this metabolic activity is caused by cross-modal plasticity due to hearing deprivation or is just the result of the physiological maturation of the cortex [63]. In fact, it was discovered that high levels of metabolic activity in the prefrontal parietal region before implantation were the strongest predictor of auditory performance with CI when confounding factors including biological age, duration of deafness, and age at implantation were controlled [67]. Although EEG has high temporal resolution, its spatial resolution is low. This technique does not necessarily correspond to the source of neural activity in the cortex because it infers cortical activity from scalp-based records, and related results should be interpreted with caution. Furthermore, the majority of previous EEG studies examining the connection between the visual takeover of auditory regions and the success of CI used visual stimuli such as checkerboard image stimulation [58,100], concentric circle stimulation [71], visual motor stimulation [70,73], and picture stimulation [72]. There are almost no studies using visual speech as a stimulus because of the shortcomings of this technique. In contrast to these abstract visual stimuli, visual speech stimuli have more cognitive components and may more accurately reflect the growth of the language network and left hemisphere specialization. Therefore, it seems to be more reasonable and ecologically valid to use visual speech as a visual stimulus. The details of those neuroimaing techniques are in part 3.

There is no consensus in research regarding the impact of the cross-modal reorganization of the auditory cortex by vision on CI success due to heterogeneity in participants, experimental techniques, and stimulus paradigm. In fact, given the small sample size and significant heterogeneity of the deaf population, it would be dangerous to draw definite conclusions and generalizations concerning the effects of cross-modal reorganization on hearing recovery. In the future, studies may need to focus on using visual speech as one type of stimulus to assess cortical function prospectively and longitudinally in much more deaf populations before and after implantation, controlling for individual confounding factors. Due to methodological challenges in applying neuroimaging methods to CI populations, such studies remain scarce. We now evaluate the use and future prospects of fNIRS, which appears to be a promising technique with potential for CI recipients.

## 3. fNIRS in Deafness and Cochlear Implant

Current brain imaging techniques include fMRI and PET–CT as well as EEG. Because they visualize intracerebral connections and allow for quantitatively examining brain networks, fMRI and PET-CT are frequently utilized to comprehend the neurological mechanisms connected to auditory rehabilitation. However, the data obtained from these imaging methods is limited are external noise and artifacts (fMRI) [98,101] and invasive radioisotope effects (PET) [102]. In addition, fMRI is incompatible with CI, which restricts post-implant longitudinal follow-up [98]. EEG is widely utilized in hearing and speech studies because it is able to clearly show the process of development and neuroplasticity of the auditory center. The main drawback of this technology is the interferences caused by the electric artifacts generated by the CI device’s electrical components. Consequently, short sounds like square-wave pulses must be employed as stimuli to get around those negative effects. Additionally, sedative medications should be used in conjunction to keep individuals calm during the EEG test, particularly in infants and young children [103,104,105].

fNIRS is an optically based technique that images the hemodynamic response to neuronal activity in the brain based on the differential absorption of near-infrared light by oxyhemoglobin (HbO) and deoxyhemoglobin (HbR). The intensity of the light that returns to the scalp’s surface after being directed through the cortex using low-power near-infrared light is measured. Changes in the concentration of HbO and HbR can be measured that are subsequently interpreted as an indirect reflection of neuronal activity [21,106,107]. Figure 1 illustrates the general working principle of fNIRS. The use of fNIRS as a neuroimaging technique has a number of benefits, including silence (which enables the examination of auditory stimulus response), portability (which enables imaging anywhere), resistance to head movement (which enables testing even when infants and toddlers are awake), compatibility with hearing aids, and no environmental interference. A comparison between fNIRS and other techniques is given in Table 2. It proves that the fNIRS technique is feasible for evaluating neural activity due to good correlations with the temporo-spatial characteristics of the fMRI signal [106,108,109]. Due to its limited spatial resolution, which falls between EEG and fMRI, fNIRS has not historically been widely used to examine the underlying brain structure. However, with the use of 3D digitizers and spatial registration software, now it is possible to infer the anatomical sources of cortical activation measured using fNIRS [110,111,112]. Above all, fNIRS technology gives CI users the chance to conduct long-term analysis of changes in their cortical function.

### 3.1. fNIRS as a Tool for Predicting/Assessing Speech Performance in Cochlear Implants

The viability of fNIRS in pediatric and adult CI recipients was originally described by Sevy et al. [106]. A 4-channel NIRS system was used in the study to measure the activation of bilateral temporal regions in response to 20-s auditory and visual stimulation. The results found that fNIRS was able to detect cortical activity in CI users even on the day of implantation and 4 months after implantation without restricting the stimulus paradigm. Since then, low-level stimulation, along with language and nonlinguistic auditory stimulation, have been administered to CI adults using fNIRS [19,113,114,115,116]. Researchers gradually attempted to analyze the functional relationship between language-induced cortical activity and auditory performance after the feasibility and effectiveness of fNIRS in CI users were confirmed. In 35 postlingual CI individuals, Olds et al. [19] used fNIRS measures of speech-evoked brain activation to examine the neurological correlates of speech comprehension. The study showed that the cortical activity of CI adults with good speech perception was similar to that of normal adults; that is, both of them exhibited greater cortical activation for natural speech than for unintelligible speech. In contrast, CI adults with poor speech perception showed the nondifferential activation of all kinds of stimuli. It is encouraging for future applications of fNIRS to objectively test speech understanding in clinic and to comprehend the neurological substrate of varying CI outcomes [19,98,117,118].

The assessment and evaluation of the plasticity of the auditory cortex both before and after implantation is another practical application of fNIRS. In a fNIRS study, Dewey and Hartley [119] explored the activation of the temporal cortex in response to auditory and visual stimuli in 30 profoundly deaf participants and 30 normally hearing controls. They found that compared with controls, visual stimuli induced a stronger response in the right temporal lobe in deaf subjects, which confirmed that the cross-modal reorganization of the auditory cortex by vision is related to auditory deprivation and that fNIRS is able to detect the process of neuroplasticity. Chen et al. [114] employed fNIRS to investigate cross-modal reorganization in the auditory cortex and the relationship between this reorganization and post-implant hearing rehabilitation. The study found that the variability of speech rehabilitation in CI users depended on the combined effect of cross-modal reorganization in the auditory cortex and visual cortex. In addition, the reorganization in the auditory cortex was detrimental to CI recovery, while the reorganization in the visual cortex was beneficial to CI recovery. These results support the previous conclusion that the cross-modal reorganization of the auditory cortex by visual stimulation plays a crucial role in auditory outcomes after CI. Anderson et al. [93] used the difference in cross-modal reorganization in the auditory cortex pre-implantation to predict the auditory performance after CI. The results demonstrated that the stronger the response to visual speech in the auditory cortex before CI, the worse the speech intelligibility after CI. However, further analysis found that this relationship was caused by individual differences among subjects. Even so, preoperative functional imaging of the auditory field provided higher prognostic value than influential clinical characteristics, including age at onset and duration of hearing deprivation, indicating that preoperative fNIRS imaging could objectively assess the physiological state of the brain and accurately predict CI outcomes. These same subjects were followed from the day of implantation to 6 months after implantation to observe the changes in cortical activation to visual speech in temporal regions [62]. It was found that participants who had recently gone deaf had increased cross-modal activity from pre- to postimplantation, while those who had been deaf for a longer period of time showed decreased cross-modal activation [62]. Furthermore, this increase in cross-modal reorganization was positively correlated with speech perception 6 months after CI. There was a claim that visual language’s cross-activation of the auditory cortex did not impair the cortical sensitivity to auditory stimuli and that there may even be a synergistic relationship between visual and auditory language [62,64]. According to those authors’ conclusions, the visual take-over of the auditory brain may therefore give adaptive benefits for hearing recovery following implantation as opposed to maladaptive effects [62,64]. Table 3 summarizes the milestone studies on the use of fNIRS for neuroimaging, including the key findings, advantages, limitations, authors, and published year.

The safety and test–retest reliability of fNIRS also make it a good candidate for investigating the cortical mechanisms associated with neuroplasticity. For instance, understanding the cortical reorganization occurring in CI candidates after prolonged auditory deprivation may help predict auditory outcomes after implantation. Additionally, fNIRS could also provide longitudinal follow-ups of cortical changes in deaf patients after implantation. One example of such an application is studying post-implantation training and its effect on cortical plasticity and eventually guiding the design of auditory and language rehabilitation strategies.

### 3.2. Future fNIRS Application Directions in CI Users

The basic goal of CI is to improve a person’s capacity for speech recognition and discrimination. In fact, it is necessary to adjust and program the CI repeatedly to ensure that the acoustic information delivered by the cochlear implant can effectively stimulate the auditory nerve and the corresponding cortex. However, this adjustment and programming of CI depend on behavioral feedback in speech perception tests due to the limitations of assessment tools, which are often unreliable, especially with infants and toddlers. An objective method used to measure whether or not auditory input is successfully processed in the temporal region will be valuable for monitoring and predicting the language development and auditory rehabilitation of CI users. If the auditory and language areas of the brain are properly activated, CI users may be in the best state for hearing and learning speech. On the other hand, if the corresponding brain regions are not activated, it implies that immediate interventions and adjustments are necessary to prevent the delay of language and psychosocial development. We proved that fNIRS is an effective imaging method for assessing cortical responses in CI recipients. There is a substantial body of research demonstrating the ability of fNIRS to objectively assess language intelligibility [19,117,118], the hemispherical lateralization of auditory response [120,121,122], and the degree of listening effort at the cortical level [117,118,123]. These findings lay the foundation for future clinical applications in the assessment of speech perception in CI recipients.

In fact, to apply individualized therapy and advance the language development of deaf groups, precise prediction of the trajectory of speech and language capacity after implantation will be the first step. There is a great deal of variability in hearing recovery even in early implanted CIs. Whether children will develop language abilities that are age-appropriate or will continue to have persistent language delays cannot be predicted in a practical way. We have reviewed that the cross-modal reorganization of the auditory cortex by vision plays an important role in predicting the prognosis of CI. Monitoring and predicting CI outcomes in real time is now possible with longitudinal follow-up investigations of cortical activation before and after implantation using fNIRS, and this objective method could easily be standardized across clinical centers [124]. Future multicenter clinical studies can be carried out to establish regression functions and thereby estimate variability between CI individuals. These functions will subsequently be helpful for assessing the chances of a patient’s recovery. Additionally, CI users can receive customized and directed training based on these predictions to maximize their auditory and speech rehabilitation.

## 4. Conclusions

Although CI could significantly improve speech comprehension in patients with severe-to-profound sensorineural deafness, the results are highly variable and unpredictable [11,125]. Disclosing the factors and mechanisms behind this variability in CI outcomes is of clinical importance because it is beneficial for guiding and adjusting rehabilitation measures for CI users. The fact that the central auditory system appears to adapt to hearing loss first before adapting to the restoration of auditory input shows how important the central function is in this area [126]. A growing body of evidence suggests that the temporal cortex may be an important element in comprehending and predicting CI consequences [58,70,92,114,127]. In this study, we aimed to discuss both the potential adaptive and detrimental implications of cross-modal reorganization in the auditory cortex on hearing recovery and possible reasons for that. From the existing evidence, we believe that it may be necessary to distinguish between the plasticity-promoting and maladaptive effecta caused by different visual stimulation signals (low-level visual stimulation and high-level visual language stimulation), as well as the various effects brought on by the reorganization of the left and right temporal cortices, even though these are not yet fully understood. We should concentrate on the connections between cortical activity caused by high-level visual language stimuli (sentences) and CI recovery in the future because the ultimate goal of CI is for nonhearing individuals to be able to integrate into society and have normal conversations as sentences involve many contextual and predictive clues, distinct syntactic and prosodic features, and more.

In addition, how does the auditory cortex change after obtaining auditory signals again after undergoing cross-modal development? Due to the limitations of technical solutions, this question has not yet been resolved. Established neuroimaging methods, including fMRI, EEG, MEG, and PET, have limited compatibility with CI devices or are not generally appropriate for longitudinal research due to safety concerns [101,104]. These drawbacks appear to be avoidable by using the neuroimaging technology known as fNIRS [21,107]. Relevant reports have confirmed the exciting results of fNIRS in the study of cortical functional plasticity in deaf and CI users. In particular, Anderson et al. [62] proved the feasibility of fNIRS in a longitudinal study of the cortical functional changes in CI users, although there were individual differences in the experiment. In conclusion, fNIRS provides a powerful tool for identifying cortical function, encouraging clinicians and audiologists to comprehend the primary mechanisms of auditory recovery, and ultimately achieving the goal of assessing and predicting CI outcomes.

## Figures and Tables

**Figure 1 brainsci-12-01150-f001:**
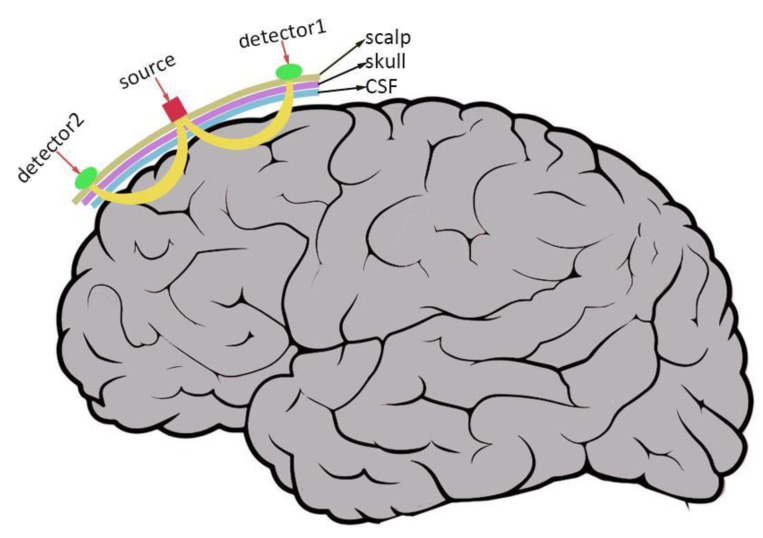
The general working principle of fNIRS: The light source (“source”) emits a mixture of near-infrared light wavelengths that are directed at the surface of the head. The photodetector (“detector”) on the scalp detects photons returning to the surface of the skull after taking a banana-shaped path (in yellow) in the tissues. The number and configuration of the detector and source on the head varies between studies. CSF stands for cerebrospinal fluid.

**Table 2 brainsci-12-01150-t002:** Comparison between fNIRS and other techniques.

Techniques Comparison	fNIRS	EEG	fMRI	PET
Spatial resolution	Medium	Low	High	High
Temporal resolution	Medium	High	Low	Low
Confounding effect of techniques	No	No	Yes, because of high levels of scanner noise	No
CI-compatible	Yes	Yes	No	Yes
Interference of CI	No	Yes, electric artefacts created by the device.	Yes, magnetic artefacts created by the device.	Yes, metal artefacts created by the device.
Safety for repeated testing	Yes	Yes	Yes	No, because of radionuclide exposure
Invasiveness	Noninvasive	Noninvasive	Noninvasive	Invasive
Physical constraints of subjects	Low, resistant to movement artefacts	Medium	High, susceptible to movement artefacts	High, susceptible to movement artefacts
Running cost	Low	Low	High	High
Ecological validity	High	Medium	Low	Low
Portability	Yes	Yes	No	No
Depth of detection	Surface of the cortex	cortex	Deep nuclei	Deep nuclei

**Table 3 brainsci-12-01150-t003:** Key findings and advantages in the milestone studies on the use of fNIRS.

Manuscripts	Key Finding	Advantages	Limitations
Sevy et al., 2010 [106]	NIRS has the potential to compensate for the shortcomings of behavioral assessment tools by providing an accurate measure of a CI’s ability to successfully stimulate the auditory cortex.	A seminal study that demonstrates the feasibility of NIRS neuroimaging in pediatric and adult CI recipients.	This is a preliminary study that does not look into whether there is a difference in cortical response between standardized speech testing materials in the clinic and the stimuli used in the article or what features of the acoustic stimulus are the most effective drivers of the NIRS response.
Dewey et al., 2015 [119]	Profoundly deaf individuals show increased activation of visual stimulation in the right auditory cortex compared with normal-hearing controls using fNIRS. There is no significant difference in activation to somatosensory stimulation between groups.	This is the first study to report cross-modal cortical responses in profoundly deaf individuals, and it demonstrated the potential of fNIRS for studying cross-modal cortical plasticity prior to and following cochlear implantation in all age groups.	Due to the heterogeneity of the deaf group and the imbalance in sample size between prelingually deaf individuals and postlingually deaf individuals, it was not possible to perform extensive subgroup analyses.
Olds et al., 2016 [19]	Implanted adults with good speech perception and NH controls show a similar pattern, that is, greater cortical activation for natural speech than for unintelligible speech. Poor CI users have indistinguishable cortical activation for all stimuli. Although CI participants’ cortical activation directly correlates with the CNC (R^2^ = 0.53 to 0.68) and AzBio (R^2^ = 0.55 to 0.66) scores, it does not correlate with their general auditory abilities (SRT scores).	The study reveals the neural correlates of speech processing among CI adults so that the variability in CI outcomes can be better understood. Therefore, fNIRS could be used as an objective measure of speech perception.	This study does not disclose how the general auditory abilities (SRT scores) were measured and does not quantify or control the attention of the participants.
Chen et al., 2016 [114]	The variability in speech rehabilitation in CI users depends on the combined effect of cross-modal reorganization in auditory cortex and visual cortex. In addition, the reorganization in auditory cortex is detrimental to CI recovery, while the reorganization in visual cortex is beneficial to CI recovery.	This is the first fNIRS study to investigate the joint influence of functional reorganization of both auditory and visual cortex on CI users’ speech recognition, and the results indicate the importance of both types of reorganization.	The author does not examine the relationship between cross-modal plasticity and CI outcome while controlling confounding factors, such as age at onset and duration of auditory deprivation.
Anderson et al., 2017 [62]	The increased cross-modal activation of auditory brain regions by visual speech from before to after implantation is adaptive to hearing restoration after implantation through an audiovisual mechanism. Furthermore, there is a strong positive correlation between changes in bilateral STC activation to visual speech from preimplantation to postimplantation and speech understanding with a CI (r = 0.70).	A longitudinal study that used fNIRS to examine changes in cortical function and plasticity over the period from hearing loss to hearing rehabilitation with a CI.	There was heterogeneity in these subjects, who include prelingually, perilingually, and postlingually deaf individuals, so incorporating these subjects into a unified framework of analysis requires care.
Anderson et al., 2019 [93]	fNIRS measures can provide additional prognostic information about future CI outcomes. Preoperative cortical imaging provides prognostic value above that of influential clinical characteristics, including the age at onset (an additional 18%) and duration of auditory deprivation (an additional 35%).	The study suggests that the use of fNIRS as an objective measure prior to cochlear implantation may enable us to deliver more accurate prognostic information to adult CI candidates.	The relationship between cross-modal plasticity and auditory outcomes was driven by the heterogeneity in adult CI-using clinical populations.
Mushtaq et al., 2020 [64]	Although CI users display significantly greater cortical responses to visual speech compared with NH controls, there is no significant difference between these two groups in responses to auditory speech. Visual and auditory speech are processed synergistically in the temporal cortex of children with CIs, and they should be encouraged, rather than discouraged, to use visual speech.	The first fNIRS study with pediatric CI recipients explores the relationship between speech understanding and cortical responses.	Almost all CI users score well on the speech perception test, so the authors do not compare the differences in cortical responses between CI users with good vs. poor speech perception.

## Data Availability

Not applicable.

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
