# Peer review of "The Effects of Cortical Reorganization and Applications of Functional Near-Infrared Spectroscopy in Deaf People and Cochlear Implant Users"

_brainsci, 2022, doi:10.3390/brainsci12091150_

Round 1
Reviewer 1 Report
In this paper, the authors emphasize the importance of cortical factors, namely the auditory cortex reorganization. The review provided is pretty comprehensive and their work clarifies the viability of using fNIRS (Functional Near-Infrared Spectroscopy) as a neuroimaging tool to predict and evaluate speech function in Cochlear Implant (CI) recipients. The topic addressed by the current paper is of great importance, as its results aim to encourage clinicians and audiologists to comprehend the primary mechanisms of auditory recovery, and ultimately achieve the goal of assessing and predicting CI outcomes.
Overall, the manuscript is well written, the use of English language is very good (some small errors cna be found, as well as some typos but nothing of great concern) and the sections follow clearly, giving a good overview of the matter discussed. Although it is a lengthy paper, the thoroughness of it justifies the length. The results follow in line with the argumentation presented in this paper, making a valuable contribution to knowledge and understanding of the factors and mechanisms behind the variability in CI outcomes, especially the ones related to the potential adaptive and detrimental implications of cross-modal reorganization in the auditory cortex on hearing recovery and possible reasons for that.
Author Response
Thanks for your kind suggestion, and the errors and typos in the manuscript have been revised.
Reviewer 2 Report
The authors provide a review the litterature of cortical activation patterns in cochlear implant processes and highlight the complex situation both regarding individual differences and measurement techniques and their shortcomings.
Finally they propose the use of an optical method (fNIRS) instead as having both better resolution capacity than EEG and full usefulness in the CI operated patient (in contrast to fMRI methods), in addition to being cheap and simple.
What I miss in the paper is a methods description with graphics and examples of data results (which do not appear so very easy to interpret?). Furthermore it would be of interest to see some preliminary examples of results in CI operasted cases. The method has been known since 1977 I saw in Karim et al, Gait Posture 2013!
Author Response
Thanks for your kind suggestion. Just like you said, a methods description with graphics and examples of data results is preferable to interpret the topic in the review. Since it is challenging to obtain raw data and the permission to use figures from the authors, we added Table 1 to list the major studies on the correlations between cross-modal reorganization and CI outcomes. This Table includes the authors, year, study population, stimuli, neuroimaging method, main findings, and study design. Please see the attachment.

Reviewer 3 Report
The authors reviewed recent studies on auditory cortex reorganization in deaf patients and cochlear implants and then discussed the feasibility of using Functional Near-Infrared Spectroscopy as a neuroimaging tool in assessing speech performance in cochlear implant recipients. However, previous work has not been well organized and comprehensively described. Here are some comments to the authors:
1. Please list a table summarizing the milestone studies on the use of fNIRS for neuroimaging, including the key findings, advantages, limitations, authors, and published year.
2. Please use a table or any illustration to show a comprehensive comparison between fNIRS and other techniques, highlighting the major differences/improvements of fNIRS and the importance and benefit of using fNIRS.
3. As fNIRS is the focus of this review, the authors should at least include some diagram or illustration to inform the readers of the general working principle.
Author Response
Thanks for your kind advice. We have added i) table 3 to summarize the the milestone studies on the use of fNIRS for neuroimaging, including the key findings, advantages, limitations, authors, and published year; ii) table 2 to show a comprehensive comparison between fNIRS and other techniques; iii) Figure1 to illustrate the general working principle of fNIRS. Please see the attachment.
